# Nimble: Lightweight and Parallel GPU Task Scheduling for Deep Learning

**Woosuk Kwon,** **Gyeong-In Yu,** **Eunji Jeong, Byung-Gon Chun**$^{†}$
Seoul National University
{kws9603,gyeongin,ejjeong,bgchun}@snu.ac.kr

## Abstract

Deep learning (DL) frameworks take advantage of GPUs to improve the speed of DL inference and training. Ideally, DL frameworks should be able to fully utilize the computation power of GPUs such that the running time depends on the amount of computation assigned to GPUs. Yet, we observe that in scheduling GPU tasks, existing DL frameworks suffer from inefficiencies such as large scheduling overhead and unnecessary serial execution. To this end, we propose Nimble, a DL execution engine that runs GPU tasks in parallel with minimal scheduling overhead. Nimble introduces a novel technique called ahead-of-time (AoT) scheduling. Here, the scheduling procedure finishes *before* executing the GPU kernel, thereby removing most of the scheduling overhead during run time. Furthermore, Nimble automatically parallelizes the execution of GPU tasks by exploiting multiple GPU streams in a single GPU. Evaluation on a variety of neural networks shows that compared to PyTorch, Nimble speeds up inference and training by up to 22.34× and 3.61×, respectively. Moreover, Nimble outperforms state-of-the-art inference systems, TensorRT and TVM, by up to 2.81× and 1.70×, respectively.

## 1   Introduction

In recent years, growing demands for deep learning (DL) have facilitated the advance of DL frameworks such as Caffe2 [23], MXNet [13], PyTorch [27], and TensorFlow [9]. These frameworks provide implementations of GPU-based neural network computations along with high-level APIs, with which users can express the semantics of neural networks as usual Python programs. Furthermore, such frameworks allow users to describe the training and inference procedure of their networks without the need to control GPUs directly. DL frameworks then automatically handle GPU intricacies such as copying neural network weights to GPUs and launching DL operators on GPUs. Operators indicate numerical computations, like convolution and batch normalization, and consist of one or more *GPU tasks* (i.e., GPU kernels and GPU memory operations).

Before a GPU processes a task, DL frameworks must first go through a series of preparation steps (*GPU task scheduling*), and then submit the task to the GPU (*GPU task submission*). We note that current DL frameworks conduct GPU task scheduling during *run time*. For instance, TensorFlow, Caffe2, and MXNet represent a neural network as a computation graph of DL operators, and schedule the GPU tasks of an operator at run time once the operator's dependencies are met. Meanwhile, for PyTorch and TensorFlow Eager [10], GPU tasks are scheduled at run time as Python code is interpreted line by line.

---

$^{*}$First two authors have equal contribution
$^{†}$Corresponding author

While under ideal circumstances the running time of neural networks mostly depends on the amount of computation assigned to GPUs, in reality we find otherwise. We point out two important problems in run-time task scheduling that may significantly limit framework performance. First, the time spent on scheduling, which we call *scheduling overhead*, can take a substantial portion of the overall running time. Although the scheduling overhead is negligible when the running time of a GPU task is sufficiently long enough to hide the overhead, we find that this does not hold in many cases, especially when inference and training of a neural network consist of small and short GPU tasks. Modern GPUs [2, 4] have thousands of computation units along with specialized processors like Tensor Core [5], and use high bandwidth memory [16] to avoid bottlenecks from memory bandwidth. While the time spent on running GPU tasks can dramatically be reduced by such GPUs, we observe that the scheduling overhead is constantly imposed by every GPU task, and often dominates the running time of DL inference and training.

Another problem DL frameworks face is that serial execution of GPU tasks misses the opportunity to further improve performance by parallelizing task execution. Recent neural networks exhibit inter-operator level parallelism. For example, topologies of the neural networks obtained by neural architecture search (NAS) [11, 12, 25, 28, 29, 39] are directed acyclic graphs (DAGs) with multiple branches rather than linear chains. In addition, recent works have proposed new types of layers that consist of smaller operators arranged in parallel, such as MixConv [35] and Split-Attention [38] blocks. Leveraging inter-operator parallelism can lead to performance improvements in executing such neural networks, especially in the case of inference. However, existing DL frameworks [9, 13, 27] are designed and optimized to schedule GPU tasks to be executed one at a time, and thus hardly exploit inter-operator parallelism.

To address the above limitations, we present Nimble, a new DL execution engine that schedules GPU tasks to run in parallel with minimal scheduling overhead. The key observation that drives the design of Nimble is that for static neural networks the behavior of a network is predetermined by its architecture. For both inference and training, DL frameworks run the exact same computation graph with the same shapes of inputs over and over again. Thus, we can leverage detailed information about the computation graph and the input shape to optimize the scheduling of GPU tasks.

To avoid the scheduling overhead, Nimble introduces a novel *ahead-of-time (AoT) scheduling* technique. Nimble schedules GPU tasks for a given neural network execution ahead of time; later when Nimble is given an input, Nimble skips scheduling and proceeds immediately to task submission. Since the preparation steps of GPU tasks are invariant to each neural network execution (i.e., independent of the input values), we only need to perform task scheduling once. While Nimble's AoT scheduler performs GPU task scheduling, it records a trace of GPU tasks and GPU memory requests, and generates a *task schedule*. The task schedule contains all information and resources (i.e., result of the scheduling) required for the execution of the neural network, including the submission order between GPU tasks, function arguments for the GPU tasks, and how to run GPU tasks in parallel. At run time, Nimble substitutes the high-overhead scheduling procedure by the raw submission of GPU tasks based on the task schedule, dramatically reducing the scheduling overhead.

To execute multiple GPU tasks in parallel on a GPU, Nimble employs *automatic multi-stream execution*. Although the CUDA programming interface provides Stream API for concurrent kernel execution [1], assigning neural network operators to appropriate streams is a difficult task for users. Nimble automates the stream assignment and synchronization process. Before AoT scheduling, Nimble analyzes dependency relationships between operators and finds an optimal stream assignment that guarantees the smallest number of synchronizations across streams while parallelizing as many operators as possible. Given the operator-to-stream mapping, Nimble rewrites the computation graph of the given neural network to run the GPU tasks of the operators on their corresponding streams with proper synchronizations. The modified graph is then used as an input to the AoT scheduler, which in turn embeds the information about the stream mapping and synchronization in the task schedule.

Nimble is built on top of PyTorch and supports both inference and training of neural networks. Users can seamlessly apply Nimble to their PyTorch programs by wrapping DL model instances in Nimble objects. Our evaluation on a variety of deep neural networks shows that Nimble improves the speed of inference and training by up to 22.34× and 3.61× compared to PyTorch, respectively. Moreover, Nimble outperforms state-of-the-art inference systems, TensorRT [3] and TVM [14], by up to 2.81× and 1.70×, respectively. Nimble is publicly available at `https://github.com/snuspl/nimble`.

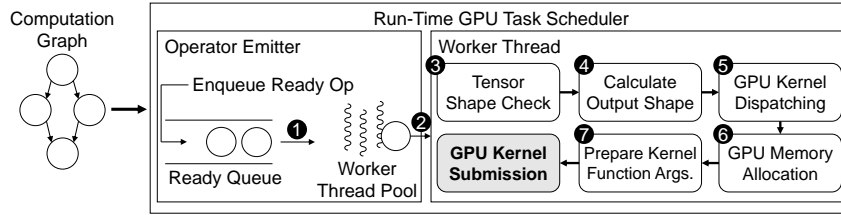

Figure 1: GPU task scheduling in DL frameworks that build a computation graph for DL execution.

## 2 Background

Here, we provide background on GPU task scheduling in existing frameworks and GPU streams.

**GPU Task Scheduling in DL Frameworks** The task scheduling mechanisms of existing DL frameworks are largely divided into two categories. First, DL frameworks including TensorFlow, Caffe2 and TorchScript [8] express a neural network as a computation graph where each node represents a DL operator and each edge indicates a dependency between two operators. The runtime stack of such a DL framework consists of two major system components (written in C++): the operator emitter and the workers. The operator emitter maintains a queue of operators whose dependencies are met and emits the operator at the front of the queue to a worker thread. The worker takes the emitted operator and performs a series of preparation steps and finally submits GPU kernels for each operator. As such, DL frameworks in this category *schedule* the GPU tasks at run time through the interplay of the operator emitter and the workers.

Second, DL frameworks including PyTorch and TensorFlow Eager describe a neural network as an imperative Python program. In such DL frameworks, there is no explicit computation graph of the neural network nor operator emitter in the runtime stack. That is, the operators are emitted by the Python interpreter as the program is executed line by line. The emitted operators are then processed by the worker in a similar manner to the DL frameworks in the first category. As such, DL frameworks in the second category also perform the run-time scheduling of GPU tasks, through the Python interpreter and the worker.

Figure 1 illustrates in detail how DL frameworks such as TensorFlow and Caffe2 carry out run-time scheduling. To submit a GPU task, the run-time scheduler must go through the following process: ❶ select an operator from the ready queue; ❷ emit the operator to a vacant worker thread; ❸ check the types and shapes of input tensors; ❹ calculate the types and shapes of output tensors; ❺ dispatch appropriate GPU kernels for the operator based on tensor types and shapes; ❻ allocate GPU memory for the output tensors and workspace for the kernels, typically by retrieving memory blocks from the cached pool of GPU memory; and ❼ prepare function arguments required for submitting the kernels. While specific steps may differ across DL frameworks, the overall process remains the same.

**GPU Streams** GPUs provide high throughput in tensor computation due to their capability to run thousands of threads in parallel. To fully utilize the computation power of a GPU, GPU kernels must have a sufficient level of intra-kernel parallelism [1]. Unfortunately, this is not always possible because the number of threads is often limited by various factors, including the implementation of the kernel and the size of the tensor being computed. Another way to enhance the GPU utilization is to schedule multiple GPU tasks to run in parallel using multiple GPU streams. A GPU stream is a queue of GPU tasks where the tasks are scheduled sequentially in FIFO order. While kernels on the same stream cannot be executed concurrently, kernels on different streams can be computed in parallel, occupying different parts of the GPU resources. The execution order between them, however, is not guaranteed unless explicitly specified by stream synchronization primitives [1]. Note that existing DL frameworks are designed and optimized to submit GPU kernels to a single GPU stream. For example, TensorFlow uses a single *compute stream* per GPU for running its kernels.

## 3 Motivation

In this section we present experiments describing the problems in GPU task scheduling of current DL frameworks. The experiments are conducted on TensorFlow [9] and PyTorch [27], the two most popular DL frameworks. The experiment setting is the same as that of the evaluation in Section 5.

**High Scheduling Overhead Makes GPUs Idle** We experimentally demonstrate that the run-time scheduling often incurs prohibitive amount of scheduling overhead such that GPU idle time dominates overall running time of DL execution. Figure 2a shows the ratios of the *GPU active time*, sum of the

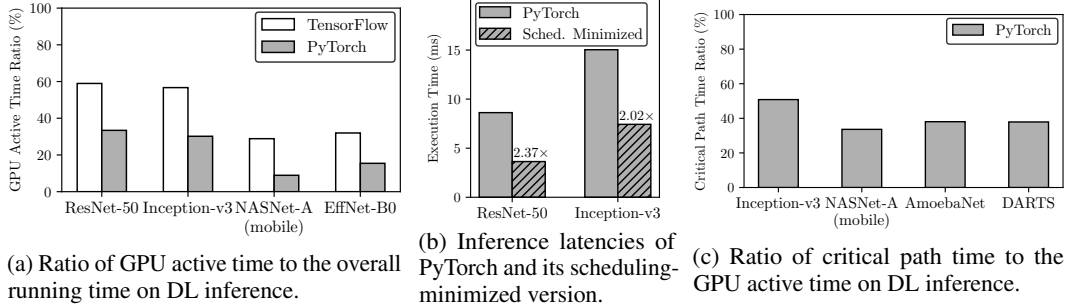

(a) Ratio of GPU active time to the overall running time on DL inference.

(b) Inference latencies of PyTorch and its scheduling-minimized version.

(c) Ratio of critical path time to the GPU active time on DL inference.

Figure 2: Various experiments showing inefficiencies in existing DL frameworks regarding scheduling overhead and serial execution.

time intervals during which GPU is not idle, to the overall running time spent on the inference of the neural networks [21, 33, 34, 39] with batch size 1. In the result, both TensorFlow and PyTorch leave their GPUs idle for a substantial portion of the running time, up to 71% and 91%, respectively. While the inefficiency in PyTorch can be partially attributed to the slowness of Python interpreter, the high overhead in TensorFlow implies that the major source of the performance bottleneck lies in the core runtime stack of the framework, and that the overhead remains significant even if the runtime is written in low-overhead language such as C++.

To further support our idea, we measure the performance of a DL framework when its scheduling procedure is minimized. For the experiment, we write a C++ program that can only perform the inference of the specific neural networks [21, 33] with a fixed input shape and uses the same GPU kernels and memory operations as PyTorch. From the assumptions that the given neural network is static and the shape of its input tensor is fixed, we prune away any redundant routines that can be done ahead of the run time. For example, shape check is omitted and the shapes of the output tensors are hardcoded in the program since every shape information can be inferred ahead of time based on the neural network architecture and the predetermined input shape. In this way, the program directly submits the GPU kernels at run time without going through the PyTorch's runtime stack for dispatching them. Likewise, GPU memory allocation is skipped and the tasks reuse fixed, pre-allocated memory regions for every iteration whose addresses are also hardcoded in the program.

Figure 2b shows the impact of such optimizations on the scheduling procedure. Despite the fact that exactly the same set of GPU kernels are computed, PyTorch and its scheduling-minimized version present remarkably different inference latencies: 2.37× speedup is obtained in ResNet-50 by the simple minimization of the scheduling procedure. The result confirms that the main source of the GPU idle time is the overhead of the scheduling procedure described in Section 2. Greater performance gain is expected in those neural networks with lower GPU active time ratio (e.g., EfficientNet-B0).

**Non-Parallel GPU Task Execution**  Framework performance can be further improved by parallelizing GPU tasks. Figure 2c shows the ratios of *critical path time* to the GPU active time in the inference of the neural networks [25, 29, 33, 39] with batch size 1. The critical path time is sum of the GPU active times spent on the operators in the longest path (in terms of time) of the computation graph. The result implies that inference latency can be reduced by up to 3× when the GPU tasks are fully parallelized and executed on a sufficiently powerful GPU (i.e., a GPU that can compute every concurrent kernel simultaneously).

In spite of the potential performance gain, existing DL frameworks do not effectively support the use of multiple GPU streams. One major obstacle we found is that the high scheduling overhead significantly decreases the chance that GPU tasks on different streams are computed in parallel. For example, Figure 3 illustrates the timeline where GPU tasks A and B are scheduled in different streams. Contrary to the expec-

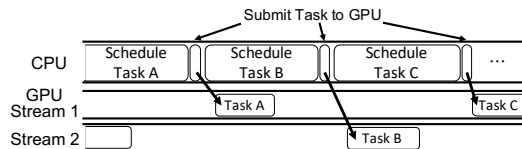

Figure 3: High scheduling overhead inhibits efficient use of multiple GPU streams.

tation that the two tasks are processed at the same time, the scheduling overhead creates a gap between the start time of the two tasks, which is longer than the duration of GPU task A. As a result, the GPU ends up executing the tasks one at a time.

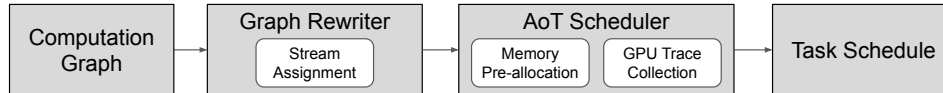

Figure 4: System overview of Nimble.

## 4 System Design

Motivated by the observations in Section 3, we present Nimble, a DL execution engine to *automatically* avoid the scheduling overhead of DL frameworks and parallelize GPU tasks using multiple GPU streams. Nimble takes a DL framework as its base system, and resolves the inefficiencies in GPU task scheduling without redesigning the framework runtime. In the current implementation, Nimble is built on top of PyTorch, but the system design is applicable to other DL frameworks.

Figure 4 summarizes execution steps in Nimble. The system consists of Graph Rewriter and AoT Scheduler. Nimble first takes as input a computation graph of a neural network. The computation graph is represented as a TorchScript [8] graph in PyTorch. The graph rewriter analyzes the computation graph and constructs an operator-to-stream mapping by the algorithm we present in Section 4.2. It marks each operator with the stream that the operator will be issued on and embeds synchronization routines to the graph by using custom nodes we add. The AoT scheduler of Nimble then goes through a series of preparation steps for the execution of the GPU tasks ahead of time. During the process, the scheduler collects a *GPU execution trace* and reserves GPU memory used in executing the GPU tasks. Finally, Nimble packs the GPU trace and the reserved memory into a task schedule. At run time, the recorded GPU tasks are replayed on the basis of the task schedule for every DL execution.

### 4.1 Ahead-of-time (AoT) Scheduling

The AoT scheduler aims to generate a task schedule, finishing the scheduling procedure required for submitting GPU tasks ahead of time. Our observation is that we can move the GPU task scheduling outside the run time execution loop without changing the semantics of neural network execution, similar to the loop-invariant code motion in compilers. In other words, while the existing frameworks repeat the scheduling procedure at every neural network execution, Nimble's AoT scheduler finishes the scheduling once ahead of time, providing a significant speedup in executing the neural network. This is possible because Nimble assumes a static neural network that performs the same set of computations for different runs, which means we can reuse the work done for scheduling after it is done once. However, this AoT scheduling raises two challenges: (a) how to distinguish the scheduling procedure that can be safely removed from the run time execution; and (b) how to move the scheduling procedure out of the run time execution.

We solve these challenges by approaching the problem from a direction different from typical performance bottleneck optimization. Instead of differentiating and removing the scheduling procedure from the run time execution, Nimble identifies *non*-scheduling work, i.e., the GPU tasks. That is, Nimble takes advantage of the fact that the computation of a given static neural network is fully described by a fixed set of GPU tasks, and that the scheduling procedure of DL frameworks becomes redundant once the set of the GPU tasks are determined. During the AoT scheduling, Nimble *pre-runs* the given neural network once according to the generated stream mapping, and records all the GPU tasks as an execution trace. The pre-run process is a single iteration of inference/training execution of the given neural network using the base framework of Nimble. During the pre-run process, while the scheduling procedure of the base framework is done as usual, the GPU tasks submitted from the framework are intercepted and recorded. The generated execution trace contains all the essential information resulted from the scheduling: dispatched GPU kernels, function arguments of the kernels, task submission order, task-to-stream assignment, etc. Once the pre-run process is done, Nimble can leverage the execution trace for submitting the tasks to the GPU, skipping the scheduling procedure.

To execute the collected GPU tasks, GPU memory should be allocated for inputs and outputs of the tasks. Since a static neural network makes the same sequence of memory requests for different runs, we can pre-allocate the exact amount of GPU memory required for its execution. For this purpose, during the process of pre-run, Nimble also intercepts memory allocate/free requests from the base framework and reserves the GPU memory allocated for the pre-run. The reserved memory is then used for the run time execution of Nimble.

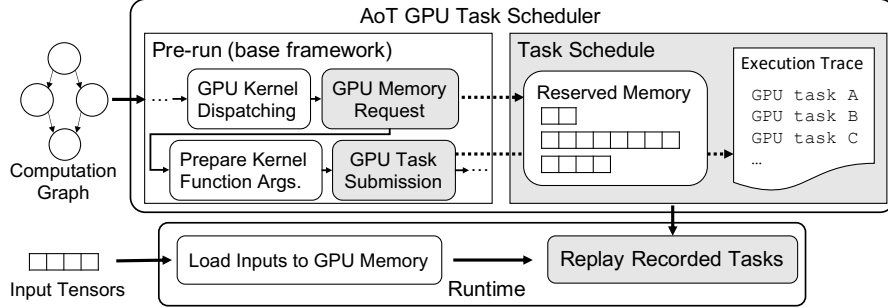

Figure 5: AoT GPU task scheduler and Runtime of Nimble. Dashed arrows represent the interception of GPU tasks and memory requests by the AoT scheduler.

At the end of the AoT scheduling, Nimble packs the execution trace and the reserved memory into a task schedule. At run time, Nimble conducts inference/training of the given neural network by directly submitting the GPU tasks recorded in the task schedule with the addresses of the reserved memory regions. In this manner, the GPU tasks can be executed independently of the base DL framework, without being tied up with the runtime and the memory allocation procedure of the base framework.

Figure 5 gives more details about the AoT scheduling technique. According to the stream assignment result, the AoT scheduler pre-runs the neural network once with a dummy input tensor. During the pre-run process, the scheduler intercepts invocations of GPU tasks and allocations of GPU memory, and constructs a task schedule. To be concrete, we use CUDA Stream Capture APIs for capturing information of GPU tasks issued on CUDA Streams, at the beginning and end of the pre-run. Then we instantiate a CUDA Graph [20], a feature introduced in CUDA 10, (i.e., execution trace representation in Nimble) from the captured information. At run time, when there is a request with a new input tensor, Nimble executes the neural network by replaying the recorded GPU tasks on the basis of the task schedule, avoiding the scheduling overhead. We execute the neural network by using CUDA Graph Launch APIs, which submit GPU tasks based on the information in the CUDA Graph.

## 4.2 Stream Assignment Algorithm

Nimble schedules GPU tasks to run in parallel by submitting them to multiple GPU streams in a single GPU. In this section, we describe an efficient algorithm for assigning GPU tasks to streams.

**Stream Synchronization**    Allowing concurrency requires proper synchronizations across streams. For example, assume that two independent GPU tasks A and B are given, and that another GPU task C consumes both outputs of A and B. If the three GPU tasks are submitted to a single stream (with the order of either A→B→C or B→A→C), no synchronization is needed. In contrast, if the three GPU tasks are submitted to three different streams, we should guarantee that GPU task C begins executing only after both GPU tasks A and B are completed. In CUDA, such dependencies across different streams can be enforced by using *events*, a special type of GPU tasks that can act as barriers. In the example, a desirable way to guarantee the execution order is to create and submit an event for each stream where GPU task A or B has been launched. We then call `cudaStreamWaitEvent` for each event to block the stream of GPU task C until both events are processed, which means that the execution of GPU tasks A and B have finished. We refer to issuing an event on the stream of task X and blocking the stream of task Y as a synchronization on the edge (X, Y). We count the number of synchronizations as the number of edges where synchronizations occur.

A few DL frameworks [27, 36] have high-level APIs through which programmers can create, switch, and block the streams where GPU tasks run. Nevertheless, as we pointed out in Section 3, leveraging multiple streams on these frameworks rarely yields performance enhancement due to their GPU task scheduling overheads. Additionally, even when the framework users are able to take advantage of the multi-stream execution, it remains as a significant burden for the users to assign and synchronize the streams in a safe and an efficient manner. Nimble resolves these difficulties by *automatically* parallelizing the GPU tasks. The process of parallelization and synchronization is transparent to users but it gives speedup when running neural networks with parallelizable structures.

**Goal of the Algorithm**    Given a computation graph, which is a DAG of DL operators, Nimble finds a *stream assignment*, a mapping from the node set of the computation graph to a stream set of the GPU. Nimble's stream assignment algorithm meets the following two goals:

---

**Algorithm 1:** Nimble's stream assignment algorithm.

| | |
|---|---|
| **Input** | A DAG $G = (V, E)$ where $V = \{v_1, v_2, ..., v_n\}$. |
| **Output** | A stream assignment $f : V \to S$. |

**Step 1** Obtain the minimum equivalent graph of $G$. We call this graph $G' = (V, E')$.

**Step 2** Define a bipartite graph $B = (V_1, V_2, E_B)$ where $V_1 = \{x_1, x_2, ..., x_n\}$, $V_2 = \{y_1, y_2, ..., y_n\}$, and $E_B = \{(x_i, y_j) \mid (v_i, v_j) \in E'\}$.

**Step 3** Find a maximum matching $M$ of the bipartite graph $B$.

**Step 4** Make a collection of sets $\{\{v_1\}, \{v_2\}, ..., \{v_n\}\}$. For each $(x_i, y_j) \in M$, combine the two sets that $v_i$ and $v_j$ are in. The result is a partition of $V$.

**Step 5** Construct $f : V \to S$ in such a way that $f(v_i) = f(v_j)$ iff $v_i$ and $v_j$ are included in the same set.

---

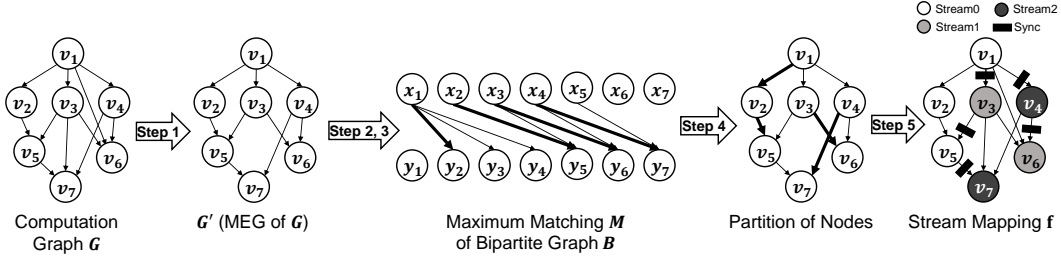

Figure 6: Example walk-through of Algorithm 1. Bold lines indicate edges in maximum matching $M$.

- **Maximum logical concurrency.** Given a neural network $G = (V, E)$ and a set of streams $S = \{s_1, s_2, ..., s_n\}$, find a mapping $f : V \to S$ such that if $x, y \in V$ and there is no dependency between $x$ and $y$ (i.e., no established order exists between the two), then $f(x) \neq f(y)$ (i.e., the two nodes are assigned to different streams).

- **Minimum number of synchronizations.** Among such functions, find $f$ that incurs the smallest number of synchronizations across streams.

Maximum logical concurrency is an optimization strategy that generalizes a common practice. To increase the chance that GPU resources are fully utilized, maximizing the concurrency is desirable. In addition, the algorithm factors in the number of synchronizations needed for safe concurrent execution. Since synchronizations hamper the fast launching of tasks, the algorithm is designed to incur the theoretically smallest number of synchronizations while maintaining maximum concurrency.

**Algorithm Description** The stream assignment algorithm of Nimble is described in Algorithm 1. Figure 6 illustrates how the algorithm is applied to a computation graph $G$. At Step 1, we compute the *minimum equivalent graph* (MEG) $G'$, which is a subgraph of the computation graph $G$ with the same set of the nodes and the smallest subset of the edges that maintains the same reachability relation as $G$. Note that the MEG of a finite DAG is unique and can be constructed in polynomial time [22]. At Step 2 and Step 3, we define a bipartite graph $B$ from $G'$ and then find a *maximum matching* of $B$, a matching that includes the largest number of edges. A maximum matching of a bipartite graph can be computed by Ford-Fulkerson algorithm [19]. At Step 4, we first create a collection of node sets where each node in the graph $G'$ is a separate set. Then for each edge $(x_i, y_j)$ in $M$, we combine the two node sets that $v_i$ and $v_j$ are in. At Step 5, nodes belonging to the same set are mapped to the same stream, and nodes belonging to different sets are mapped to different streams.

We now demonstrate that the stream assignment constructed from Algorithm 1 meets the two goals by using the following theorems. Detailed proofs on the theorems are presented in Appendix A.

**Theorem 1.** *A stream assignment $f$ satisfies maximum logical concurrency on $G$ if and only if $f$ satisfies maximum logical concurrency on $G'$. Also, for any stream assignment $f$ that satisfies maximum logical concurrency, the minimum number of synchronizations required for $f$ on $G$ is equal to the minimum number of synchronizations required for $f$ on $G'$.*

**Theorem 2.** *There exists one-to-one correspondence $\Phi$ from the set of the matchings of the bipartite graph $B$ to the set of the stream assignments that satisfy maximum logical concurrency on $G'$. In fact, $\Phi$ is constructed by Step 4 and Step 5 of Algorithm 1.*

**Theorem 3.** *For any matching $m$ of the bipartite graph $B$, the minimum number of synchronizations required for the corresponding stream assignment $\Phi(m)$ is $|E'| - |m|$.*

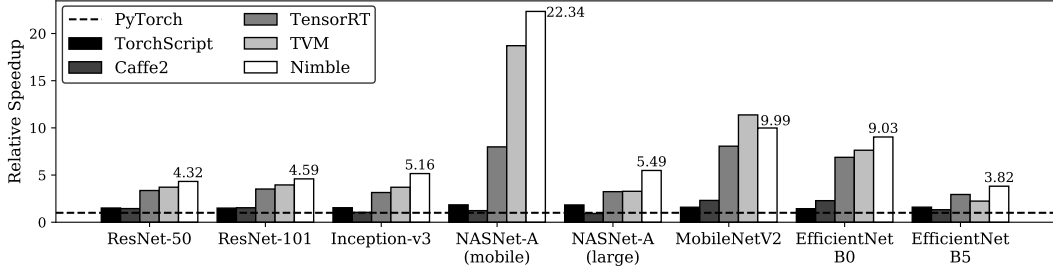

Figure 7: Relative inference speedup of Nimble and other systems (batch size 1). We use various neural networks [21, 32, 33, 34, 39], all trained on ImageNet [31].

**Theorem 4.** *For a maximum matching $M$ of the bipartite graph $B$, $\Phi(M)$ is a stream assignment that satisfies maximum logical concurrency and requires minimum number of synchronizations among the stream assignments satisfying maximum logical concurrency.*

***Proof of Theorem 4***. Based on Theorem 1, the algorithm derives the desired stream assignment from $G'$ instead of $G$. From Theorem 2, it follows that $\Phi(M)$ is a stream assignment with maximum logical concurrency. Now, suppose that there exists a stream assignment $g : V \rightarrow S$ that satisfies maximum logical concurrency with strictly less number of synchronizations than that of $\Phi(M)$. By Theorem 2 and Theorem 3, $g$ corresponds to some matching $\Phi^{-1}(g)$ of $B$ such that $|M| < |\Phi^{-1}(g)|$. The inequality, however, is contradictory to the definition of $M$ since $M$ is a maximum matching of the bipartite graph $B$. Thus, Theorem 4 follows. □

# 5 Evaluation

**Experimental Setup**   We implement Nimble on PyTorch v1.4 with CUDA 10.2 and cuDNN 8.0.2. For evaluation, we use an NVIDIA V100 GPU along with 2.10GHz Intel Xeon CPU E5-2695 v4.

To evaluate DL inference, we compare Nimble with popular DL frameworks, PyTorch, TorchScript and Caffe2, as well as state-of-the-art inference systems, TensorRT (v7.1) [3] and TVM (v0.6.1) [14]. To evaluate DL training, Nimble is compared with PyTorch and TorchScript. Note that TensorRT and TVM employ graph optimizations (e.g., aggressive operator fusion) and kernel selection/tuning, which are orthogonol to our idea. In Nimble, we also implement the operator fusion (a subset of TensorRT's) and basic kernel selection, which chooses the faster implementation of convolution operators between cuDNN [15] and PyTorch's native implementation. More details on the evaluation setting are provided in Appendix B.

## 5.1 Inference Latency

Figure 7 presents the relative inference speed of Nimble and the other systems. We set PyTorch as the baseline. The result shows that Nimble outperforms PyTorch, TorchScript and Caffe2 significantly. The primary reason for this performance gap is the substantial scheduling overhead, which makes GPU idle for most of the time. In addition, since the DL frameworks hardly utilize parallelism among operators in a neural network, the performance gap widens in the neural networks with parallelizable structures like NASNet-A (mobile) (up to 22.34×). Nimble also shows higher performance than TensorRT on all of the tested neural networks, by up to 2.81× (NASNet-A (mobile)). Moreover, Nimble surpasses performance of TVM in most cases, by up to 1.70× (EfficientNet-B5). The only exception is MobileNetV2 [32]. TVM spends two days in kernel tuning (1500 trials for each convolution), and finds much faster GPU kernels for MobileNetV2 than those of cuDNN and PyTorch. Results on different GPUs are provided in Appendix C.

## 5.2 Impact of Multi-stream Execution

We select a set of deep neural networks with parallelizable structures and investigate the impact of the multi-stream execution of Nimble on the inference latency of such neural networks. Table 1 shows the relative speedup of the multi-stream execution compared to the single-stream execution of Nimble. The result indicates that multi-stream execution of Nimble can accelerate DL inference by up to 1.88× compared to the single-stream execution, and that Nimble exploits logical concurrency to the degree (15) that programmers cannot effectively assign and synchronize the streams manually.

In addition, we observe that the acceleration rates considerably differ across the neural networks. Neural networks with a higher degree of logical concurrency tend to benefit more from the multi-

| Architecture | Speedup | Deg. | #MACs |
|---|---|---|---|
| Inception-v3 | 1.09× | 6 | 5.7B |
| DARTS | 1.37× | 7 | 0.5B |
| AmoebaNet | 1.45× | 11 | 0.5B |
| NASNet-A (M) | 1.88× | 12 | 0.6B |
| NASNet-A (L) | 1.31× | 15 | 23.9B |

Table 1: Impact of the multi-stream execution of Nimble on DL inference, compared to its single-stream counterpart. Deg. stands for maximum degree of logical concurrency of each architecture.

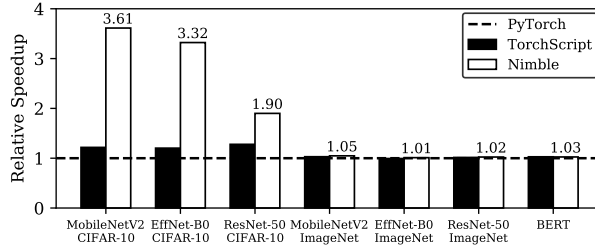

Figure 8: Relative training speedup of Nimble and TorchScript. All neural networks [18, 21, 32, 34] are trained with batch size 32.

stream execution. For example, the neural network with the lowest degree of logical concurrency (Inception-v3) gains the smallest speedups. Also, we can see the trend that neural networks with less amount of computation are more likely to be accelerated by the multi-stream execution. For instance, although NASNet-A (large) exhibits higher degree of logical concurrency than NASNet-A (mobile), the former gets limited speedup compared to the latter because the former consists of kernels with a large number of multiply-and-accumulate (MAC) operations, each of which occupies most of the GPU resources. The comparison between Inception-v3 and DARTS reports the same tendency.

### 5.3 Training Throughput

Figure 8 shows the performance of Nimble on neural network training. Since training of a neural network is commonly conducted with large batch sizes, GPU scheduling overhead imposed during training is less pronounced, and the impact of multi-stream execution is also limited. Accordingly, in the results of ResNet on the ImageNet dataset and BERT [18], Nimble shows marginal performance improvement. However, Nimble still brings up substantial speedup when neural networks are trained with small-size inputs (e.g., low-resolution images). For example, in the field of computer vision, the CIFAR-10 [24] dataset is widely used among researchers and many neural networks are trained on the dataset. Figure 8 shows Nimble's performance when neural networks [21, 32, 34] are trained on CIFAR-10. The result implies that the scheduling overhead can still be a major performance bottleneck even in training. Nimble eliminates such inefficiency and increases training throughputs by up to 3.61×. Results on different batch sizes are presented in Appendix D.

## 6 Related Works

There have been a body of works on the system-level optimization of DL inference and training. For example, DL compilers [6, 14, 17, 30, 37] have been proposed to generate optimized codes for target hardware. These works take different approach from Nimble in that they aim to reduce the time spent on GPU tasks whereas Nimble tackles the inefficiencies in the scheduling of GPU tasks.

The core ideas of Nimble can be compared with some previous works. First, in an attempt to reduce the scheduling overhead, TensorFlow recently introduced a new runtime [7] that has a thin operator dispatch routine. While redesigning a runtime stack costs tremendous engineering efforts, the AoT scheduling of Nimble provides an automated way to avoid the scheduling overhead. Second, although the pre-run process of Nimble is similar to the tracing of TorchScript, they differ in the purpose and the target of tracing process. In the tracing of TorchScript, DL operator calls are recorded to construct a computation graph, which is used for serialization and graph-level optimization. Meanwhile, Nimble records GPU tasks during the pre-run process to perform the scheduling procedure once. Lastly, in comparison to HiveMind [26] that has a parallel runtime for multi-model workloads, the multi-stream execution of Nimble parallelizes operators in a single model, using a more sophisticated algorithm.

## 7 Conclusion

We introduce Nimble, a high-speed DL execution engine for static neural networks. We first show two problems of the run-time scheduling of GPU tasks: scheduling overhead and serial execution. Nimble minimizes the scheduling overhead by finishing the scheduling procedure ahead of time before executing the GPU tasks at run time. Moreover, Nimble schedules independent GPU tasks to be executed in parallel, further boosting its performance. Our evaluation on various neural networks shows that Nimble outperforms popular DL frameworks and state-of-the-art inference systems. Nimble is publicly available at `https://github.com/snuspl/nimble`.

## Broader Impact

Our work aims to accelerate the execution of neural networks in general, and is not associated with a specific application. Furthermore, our technique does not affect the output values of neural networks (e.g., image classification labels, object detection bounding boxes, computer-generated text, etc.) nor the weights of neural networks. Therefore, we believe our work has no significant impact on any particular audience from an ethical/societal perspective, at the application-level.

## Acknowledgments

We thank the anonymous reviewers for their valuable comments. We also thank Joo Seong Jeong, Gyewon Lee, Jeongyoon Eo and Jae-Won Chung for their fruitful feedback. This work was supported by the Institute for Information & communications Technology Planning & Evaluation (IITP) grant funded by the Korea government (MSIT) (No.2015-0-00221, Development of a Unified High-Performance Stack for Diverse Big Data Analytics), the ICT R&D program of MSIT/IITP (No.2017-0-01772, Development of QA systems for Video Story Understanding to pass the Video Turing Test), and Samsung Advanced Institute of Technology.

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
