[Supplementary Material]

# Supplementary Materials for
# Nimble: Lightweight and Parallel GPU Task Scheduling for Deep Learning

## Appendix A  Proofs on the Stream Assignment Algorithm of Nimble

In this section, we provide detailed proofs on the theorems presented in Section 4.2.

**Problem Setting**  We assume that the computation graph of a neural network is given. The computation graph is represented as a finite DAG $G = (V, E)$. Also, we are given a set of GPU streams $S = \{s_1, s_2, \cdots, s_{|V|}\}$. Algorithm 1 must find a stream assignment $f : V \rightarrow S$, which satisfies the following conditions:

- **Maximum logical concurrency.** If $u, v \in V$ and there exists no path between $u$ and $v$ in $G$, then $f(u) \neq f(v)$.

- **Minimum number of synchronizations.** Among such functions, $f$ incurs the smallest number of synchronizations across streams.

Here we define important concepts and terminologies used in the following proofs.

**Definition 1.**  For a graph $G = (V, E)$, a **synchronization plan** $\Lambda \subseteq E$ is a set of edges on which synchronizations are planned to be performed (regardless of stream assignments).

**Definition 2.**  For a stream assignment $f$ on $G = (V, E)$, a synchronization plan $\Lambda \subseteq E$ is **safe** if it satisfies the following condition.

 *For any $(u, v) \in E$, $f(u) = f(v)$ or there exists a path $P \subseteq E$ from $u$ to $v$ such that $P \cap \Lambda \neq \emptyset$.*

In other words, the plan $\Lambda$ is safe when the execution order between every pair of adjacent nodes $u$ and $v$ is guaranteed: either by assigning them to the same streams or by performing a synchronization somewhere after $u$ and before $v$.

**Notation.**  We denote by $min_{sync}(G, f)$ the minimum number of synchronizations required when applying $f$ to the graph $G$. That is,

$$min_{sync}(G, f) = min\{|\Lambda| \in \mathbb{Z}_{\geq 0} \mid \Lambda \subseteq E \text{ is safe for } f \text{ on } G\}$$

### A.1  Proof of Theorem 1

Theorem 1 includes two statements, which are presented here as Theorem 1-1 and Theorem1-2, respectively.

**Theorem 1-1.**  *A stream assignment $f$ satisfies maximum logical concurrency on a computation graph $G$ if and only if $f$ satisfies maximum logical concurrency on the minimum equivalent graph $G'$.*

***Proof of Theorem 1-1***.  By definition of MEG, $G'$ has the same reachability relation as $G$. Thus, if no path exists between a pair of nodes in $G$, then there is no path between the same pair of nodes in $G'$, and vice versa. □

Prior to the proof of Theorem 1-2, we describe and prove Lemma 1 and Lemma 2.

**Lemma 1.**  *For a minimum equivalent graph $G' = (V, E')$ of $G$, if $(u, v) \in E'$, then $\{(u, v)\}$ is the only path in $G$ from $u$ to $v$.*

***Proof of Lemma 1.*** We will prove by contradiction. Suppose there is another path $P \subseteq E$ from $u$ to $v$ that goes through $w \in V$. By the definition of MEG, $G'$ must preserve reachability from $u$ to $w$ and $w$ to $v$. Consequently, removing the edge $(u, v)$ from $E'$ does not change the reachability relation. This is contradictory to the definition of MEG, because we can construct another subgraph $G^* = (V, E' \setminus \{(u, v)\})$, where the number of edges of $G^*$ is smaller than that of $G'$ while preserving the reachability relation. □

**Lemma 2.** *A synchronization plan $\Lambda \subseteq E$ is safe for a stream assignment $f$ on $G$ if and only if $\Lambda$ is safe for $f$ on $G'$.*

***Proof of Lemma 2.*** We first show that if $\Lambda$ is safe for $f$ on $G$, then $\Lambda$ is safe for $f$ on $G'$. We will prove by contradiction. Suppose $\Lambda$ is safe for $f$ on $G$ but not safe for $f$ on $G'$. Then there is an edge $(u, v) \in E'$ such that $f(u) \neq f(v)$ and $(u, v) \notin \Lambda$. Since $G'$ is the MEG of $G$ and $(u, v) \in E'$, $\{(u, v)\}$ is the only path in $G$ from $u$ to $v$ by Lemma 1. Consequently, $(u, v) \in E$ is an edge that $f(u) \neq f(v)$ and every path in $G$ from $u$ to $v$ does not include any edge in $\Lambda$, which is contradictory to the assumption that $\Lambda$ is safe for $f$ on $G$.

Next, we show that if $\Lambda$ is safe for $f$ on $G'$, then $\Lambda$ is safe for $f$ on $G$. We will prove by contradiction. Suppose $\Lambda$ is safe for $f$ on $G'$ but not safe for $f$ on $G$. Then there is an edge $(u, v) \in E \setminus E'$ such that $f(u) \neq f(v)$ and every path from $u$ to $v$ in $G$ does not include any edge in $\Lambda$. Since $(u, v) \notin E'$ and $G'$ preserves the same reachability relation as $G$, there must exist a node $w_1 \in V$ such that $(u, w_1) \in E'$ and a path from $w_1$ to $v$ exists in $G'$. As every path from $u$ to $v$ in $G$ does not include any edge in $\Lambda$, $f(u) = f(w_1)$ must hold to meet the assumption that $\Lambda$ is safe for $f$ on $G'$. Then, we have two vertices $w_1$ and $v$ such that $f(w_1) \neq f(v)$ and every path from $w_1$ to $v$ in $G$ does not include any edge in $\Lambda$. Since $G$ is a finite DAG, if we repeat this process, we end up with two vertices $w_n$ and $v$ with the following conditions: $(w_n, v) \in E'$, $f(w_n) \neq f(v)$, and $(w_n, v) \notin \Lambda$, which contradicts the assumption that $\Lambda$ is safe for $f$ on $G'$. □

**Theorem 1-2.** *For any stream assignment $f$ that satisfies maximum logical concurrency on $G$, the following equation holds.*

$$min_{sync}(G, f) = min_{sync}(G', f).$$

*That is, the minimum number of synchronizations required for $f$ on $G$ is equal to the minimum number of synchronizations required for $f$ on $G'$.*

***Proof of Theorem 1-2.*** This directly follows from Lemma 2. □

## A.2 Proof of Theorem 2

Prior to the proof of Theorem 2, we clarify the meaning of *the set of the stream assignments*. Let $F = \{f \mid f : V \to S\}$. We can define an equivalence relation $\sim$ on $F$ as follows.

*For stream assignments $g, h \in F$, $g \sim h$ if and only if $g = \sigma \circ h$ for some permutation $\sigma$ over $S$.*

Note that any permutation on $S$ does not affect the degree of logical concurrency and the number of synchronizations of a stream assignment. In other words, for stream assignments $g, h \in F$ such that $g \sim h$, it directly follows that 1) $g$ meets maximum logical concurrency if and only if $h$ meets maximum logical concurrency, and 2) $min_{sync}(G', g) = min_{sync}(G', h)$. Therefore, if two stream assignments can be converted to one another by some permutation on $S$, we do not differentiate the two stream assignments. Furthermore, we do not differentiate a stream assignment $f \in F$ from its equivalence class $[f]$, because we only consider which nodes are mapped to the same streams, but do not consider the exact value of $f$. From now on, we *identify* $[f]$, the equivalence class of $f$, as $f$.

**Remark.** The set of the stream assignments $\mathbb{F}$ is as follows.

$$\mathbb{F} = \{[f] \mid f : V \to S\}$$

**Theorem 2.** *Let $\mathbb{M}$ be the set of the matchings of the bipartite graph $B$ obtained from $G'$, and $\mathbb{F}_{max}$ be the set of the stream assignments that satisfy maximum logical concurrency on $G'$. Then one-to-one correspondence $\Phi : \mathbb{M} \to \mathbb{F}_{max}$ exists.*

***Proof of Theorem 2.*** We construct $\Phi$ according to Step 4 and Step 5 of Algorithm 1.

First, we show that $\Phi(m) \in \mathbb{F}_{max}$, i.e., $\Phi(m)$ meets maximum logical concurrency, for any matching $m \in \mathbb{M}$. We prove this by contradiction. Choose an arbitrary matching $m \in \mathbb{M}$ and suppose that $\Phi(m)$ does not satisfy maximum logical concurrency. In other words, suppose that a pair of nodes $v_i, v_j \in V$ exists such that there is no path from $v_i$ to $v_j$ in $G'$ but $\Phi(m)(v_i) = \Phi(m)(v_j)$. Since $v_i$ and $v_j$ are mapped to the same stream, it follows from Step 4 that there exists a sequence of edges $\{(x_i, y_{k_1}), (x_{k_1}, y_{k_2}), \cdots, (x_{k_l}, y_j)\} \subseteq m$. This, in turn, means that there exists a path $\{(v_i, v_{k_1}), (v_{k_1}, v_{k_2}), \cdots, (v_{k_l}, v_j)\} \subseteq E'$, which is contradictory to the assumption. Therefore, for any $m \in \mathbb{M}$, $\Phi(m)$ meets maximum logical concurrency.

Secondly, we show that $\Phi$ is injective. Again, we will prove by contradiction. Suppose that $\Phi(m_1) = \Phi(m_2)$ for some matchings $m_1 \neq m_2$. Since $m_1 \neq m_2$, there exists an edge $(x_i, y_j) \in E_B$ that is included in either of the two matchings. Without loss of generality, assume $(x_i, y_j) \in m_1$. Then the equation $\Phi(m_1)(v_i) = \Phi(m_1)(v_j)$ holds, and so does the equation $\Phi(m_2)(v_i) = \Phi(m_2)(v_j)$. The latter equation implies that there exists a sequence of edges $\{(x_i, y_{k_1}), (x_{k_1}, y_{k_2}), \cdots, (x_{k_l}, y_j)\} \subseteq m_2$. This, in turn, means that a path from $v_i$ to $v_j$ other than than edge $(v_i, v_j)$ exists in $E'$, which is contradictory to the assumption that $G'$ is the MEG of the graph $G$ by Lemma 1.

Lastly, we demonstrate that $\Phi$ is surjective. Assume that an arbitrary stream assignment $f \in \mathbb{F}_{max}$ is given. We construct $m_f \subseteq E_B$ in such a way that $(x_i, y_j) \in m_f$ if and only if $f(v_i) = f(v_j)$ and $(v_i, v_j) \in E'$. Then $\Phi(m_f) = f$ follows by definition of $\Phi$. $\qquad\qquad\square$

## A.3   Proof of Theorem 3

**Definition 3.** For a stream assignment $f$ on $G'$, we define $Q(f) \subseteq V$ as follows.
$$Q(f) = \{v \in V \mid \exists p \in V \ s.t. \ (p, v) \in E' \text{ and } f(p) = f(v)\}$$
That is, a node $v \in V$ is included in $Q(f)$ if and only if it has at least one parent node which is mapped to the same stream as $v$ by $f$.

**Definition 4.** For a stream assignment $f$ that satisfies maximum logical concurrency on $G'$, we define a function $R_f(v) : Q(f) \to V$ as follows.
$$R_f : v \mapsto p \ s.t. (p, v) \in E' \text{ and } f(p) = f(v)$$

**Lemma 3.** *The function $R_f$ is well-defined.*

***Proof of Lemma 3.*** By definition of $Q(f)$, $R_f(v)$ exists for any $v \in Q(f)$. What we have to show is the uniqueness of such $p$ for each $v$. Suppose $\exists p_1, p_2 \in V$ such that $(p_1, v), (p_2, v) \in E'$ and $f(p_1) = f(p_2)$. Since $f$ satisfies maximum logical concurrency, there is a path between $p_1$ and $p_2$. Without loss of generality, assume that there is a path from $p_1$ to $p_2$. Then $(p_1, v) \in E'$ can be removed from the MEG of $G$, which contradicts the assumption that $G'$ is MEG of $G$. $\qquad\square$

**Lemma 4.** *For a stream assignment $f$ that satisfies maximum logical concurrency on $G'$,*
$$min_{sync}(G', f) = |E'| - |Q(f)|.$$

***Proof of Lemma 4.*** We first show that $min_{sync}(G', f) \leq |E'| - |Q(f)|$. For any node $v \in Q(f)$, there exists an edge $(R_f(v), v) \in E'$. Observe that synchronization on the edge $(R_f(v), v)$ is redundant because $f(R_f(v)) = f(v)$. Thus, among all of the edges in $E'$, we can guarantee that at least $|Q(f)|$ edges do not require synchronizations.

Conversely, we show that $min_{sync}(G', f) \geq |E'| - |Q(f)|$. Let $\Lambda \in E'$ be a safe synchronization plan for $f$ on $G'$ such that $|\Lambda| = min_{sync}(G', f)$. Select an arbitrary node $v \in V$ and let $I_v \subseteq E'$ be the set of the incoming edges to $v$ in $G'$. If $v \notin Q(f)$, for any edge $e = (p, v) \in I_v$, $e \in \Lambda$. This is because, by Lemma 1, $\{e\}$ is the only path between $p$ and $v$, and, therefore, any safe synchronization plan must include the edge $e$. If $v \in Q(f)$, any edge $e \in I_v$ other than $(R_f(v), v)$ must be included in $\Lambda$. Thus, the following inequality holds.
$$min_{sync} \geq \sum_{v \notin Q(f)} |I_v| + \sum_{v \in Q(f)} (|I_v| - 1)$$
Clearly, the righthand side is equal to $|E'| - |Q(f)|$.

$\qquad\qquad\square$

**Theorem 3.** *For any matching $m \in \mathbb{M}$, the following equation holds.*

$$min_{sync}(G', \Phi(m)) = |E'| - |m|.$$

***Proof of Theorem 3.*** Let $m \in \mathbb{M}$ be a matching of the bipartite graph $B$. By Theorem 2 and Lemma 4, it suffices to show $|Q(\Phi(m))| = |m|$. For this purpose, we define a function $\Psi_m : Q(\Phi(m)) \to m$ and demonstrate that $\Psi_m$ is a bijection.

We first define a function $H : E' \to E_B$ as $H : (v_i, v_j) \mapsto (x_i, y_j)$. Since we construct the bipartite graph $B$ in the same manner as $H$, it is trivial that the function $H$ is bijective. Now we define $\Psi_m$ as

$$\Psi_m(v) = H(R_{\Phi(m)}(v), v), \quad \forall v \in Q(\Phi(m))$$

We can easily confirm that $\Psi_m$ is injective. Since $H$ is bijective, if $\Psi_m(u) = \Psi_m(v)$ then $(R_{\Phi(m)}(u), u) = (R_{\Phi(m)}(v), v)$. Thus, $u = v$ follows.

Next, we show that $\Psi_m$ is surjective. Select an arbitrary edge $(x_i, y_j) \in m$. Since $(x_i, y_j) \in E_B$, $(v_i, v_j) \in E'$. Also, by definition of $\Phi$, $\Phi(m)(v_i) = \Phi(m)(v_j)$. Thus, it follows that $v_j \in Q(\Phi(m))$ and $R_{\Phi(m)}(v_j) = v_i$. That is, the first coordinate of $\Psi_m(v_j)$ is $x_i$. In addition, from the definition of $\Psi_m$ and $H$, it is clear that the second coordinate of $\Psi_m(v_j)$ is $y_j$. To sum up, it follows that $\Psi_m(v_j) = (x_i, y_j)$, i.e., $\Psi_m$ is surjective.

Since $\Psi_m$ is a bijection between $Q(\Phi(m))$ and $m$, cardinality of the two sets are equal. $\square$

### A.4 Time Complexity Analysis

Since the computation graph $G = (V, E)$ is a finite DAG, its minimum equivalent graph can be obtained in $O(V^3)$ time [4]. To convert $G'$ into the bipartite graph $B$, Nimble computes the transitive closure of $G'$, which again takes $O(V^3)$ time. Additionally, in calculating a maximum matching of the bipartite graph $B$, Nimble uses Ford-Fulkerson method [3] which costs $O(VE)$ time. To sum up, the stream assignment algorithm of Nimble takes $O(V^3)$ time in total. Note that Nimble computes the stream assignment once before the AoT scheduling, so the time spent on Algorithm 1 is amortized over iterations. Therefore, the time spent on the stream assignment algorithm can be considered negligible.

## Appendix B  Details on Evaluation Setup

The experiments to evaluate the performance of Nimble, which are described in Section 5, use the implementations of the neural networks from various open-source repositories. We summarize the information below.

- torchvision repository[1]
    - ResNet-50, ResNet-101, Inception-v3, MobileNetV2
- Pretrained models for PyTorch repository[2]
    - NASNet-A (mobile), NASNet-A (large)
- PyTorch Image Models repository[3]
    - EfficientNet-B0, EfficientNet-B5
- Differentiable Architecture Search repository[4]
    - AmoebaNet, DARTS
- NVIDIA Deep Learning Examples repository[5]
    - BERT

(a) Results on an NVIDIA Titan RTX GPU.

(b) Results on an NVIDIA Titan Xp GPU.

Figure 1: Relative inference speedup of Nimble and other systems (batch size 1).

(a) Training with batch size 64.     (b) Training with batch size 128.     (c) Training with batch size 256.

Figure 2: Relative training speedup of Nimble and TorchScript.

Throughout the evaluation, TorchScript modules are created through PyTorch's tracing API. For Caffe2, TensorRT and TVM, PyTorch models are first converted into ONNX [1] models and then parsed by the respective parsers of the systems. For the evaluation on inference latency, we use synthetic $224 \times 224$ RGB images as inputs, except for Inception-v3, NASNet-A (large), and EfficientNet-B5. For these neural networks, the inputs are larger size images - $299 \times 299$ for Inception-v3, $331 \times 331$ for NASNet-A (large), and $456 \times 456$ for EfficientNet-B5 - following the description in the original literature [5, 7, 6]. For the evaluation on training, we use $224 \times 224$ RGB images for the ImageNet dataset, and $32 \times 32$ RGB images for the CIFAR-10 dataset. We use a sequence length of 128 in the experiments with BERT, following the setting used for pretraining in the original literature [2].

## Appendix C    Evaluation Results on Various GPUs

In addition to the evaluation results described in Section 5, we attach results on the different types of GPUs: NVIDIA Titan RTX and NVIDIA Titan Xp. We keep the other experimental settings the same. Note that we exclude TVM from this set of experiments because TVM needs to tune the kernels separately for each type of GPU for a long time. Figure 1 shows that Nimble achieves significant speedup across various GPU architectures ranging from Pascal to Turing.

## Appendix D  Evaluation Results on Different Training Batch Sizes

We also present results on the performance of Nimble when training the neural networks with varying batch sizes. We use an NVIDIA V100 GPU, following the setting described in Section 5. Figure 2 shows that Nimble can achieve performance improvement in the training of the neural networks on the CIFAR-10 dataset even when the batch size is sufficiently large.

## Footnotes

[1] https://github.com/pytorch/vision

[2] https://github.com/Cadene/pretrained-models.pytorch

[3] https://github.com/rwightman/pytorch-image-models

[4] https://github.com/quark0/darts

[5] https://github.com/NVIDIA/DeepLearningExamples