[Reviews · NeurIPS 2020]

Review 1

Summary and Contributions: This paper presents a DNN execution engine based on Pytorch that remembers CPU operations during a model run to eliminate them during later ones, reducing overhead for inference or training of DNNs with many small kernels.

Strengths: This is a well written paper overall that shows significant performance gains over PyTorch in multiple types of models. The core idea makes a lot of sense and the algorithms for deciding how to split a computation across streams look reasonable. I also appreciated that the authors implemented this in PyTorch and evaluated on three GPU types.

Weaknesses: This work is most applicable on networks with many small kernels, which may not be of broad interest in all cases. Nonetheless, it does help with training MobileNet and similar networks on desktop or server GPUs. I also feel that some parts of the paper overstate the contribution, either by only evaluating on these networks or by leaving out some optimized baselines. The biggest issues here are: - For inference, you should compare against an optimized inference runtime such as TensorRT. This will likely do better than PyTorch or Caffe2 do out of the box, even with TorchScript. A lot of production applications use TensorRT, ONNX runtime or similar systems to get the best performance possible. The 10x inference gains you claim in the intro now may be lower vs these. - For training, it was weird that the paper did not report performance on larger networks like Resnet and BERT. These results seem to only be for the mobile-optimized networks that will run a lot of small kernels. You should also quantify the result on larger networks even if it isn't better than the state of the art, to let the readers understand the limitations of this approach. - Some of the writing also sounds like it overstates findings a little bit -- for example, the intro says that everyone has focused on lowering FLOPs and that "surprisingly" the networks with many kernels do not perform that well, but kernel launch and memory allocation overheads are well known to anyone who works on GPU performance. This isn't the first paper to discover this issue. - The related work should also discuss TensorRT and other optimizing runtimes for inference. It should also talk about TorchScript in more detail since the core "record and then replay" idea in this work is similar at a high level.

Correctness: Yes, except for not testing against more optimized inference runtimes like TensorRT. I hope to see that in a camera-ready paper.

Clarity: Mostly. One issue I found is that it was hard to tell what GPUs the intro's plots were from since their captions didn't say that. It would be nice if the later plot captions also mentioned GPU type to make this easier to find.

Relation to Prior Work: The high level idea is similar to TorchScript but there are enough new pieces here that I think they represent a good advance over what's there, and they provide significant performance improvements. The related work section could use a bit more discussion of TorchScript, TensorRT and other systems that optimize for inference.

Reproducibility: Yes

Additional Feedback: I appreciated the implementation in PyTorch and the fairly extensive evaluation with multiple model types and GPU types. Feedback after response: The new experiments in the response improve the paper a lot in my opinion. I think this paper makes some good contributions that can also speed up some interesting larger models.


Review 2

Summary and Contributions: The paper proposes a lightweight execution engine for static neural networks. The basic idea is that when the neural networks are static, the work for preparing GPU operations can be done ahead of time so that the runtime overhead of the framework can be reduced. =================== The authors clarified their difference and advantage over TensorRT and TVM. The new results also show their advantage on some larger models. The idea of automatic applying optimization by capturing the GPU kernel call trace is clever, and the fact that Nimble can be used as a wrapper to PyTorch models makes it easy to use. Overall, I appreciate the authors' effort in identifying performance bottlenecks, and I think the paper makes some good contributions.

Strengths: The observation that GPU memory allocation and python runtime incur an overhead in DL models is reasonable, and the solutions proposed by the authors seem solid.

Weaknesses: The observation is trivial and has been pointed out in many previous works as the authors describe in Section 2. I appreciate the authors' effort in experimenting and identifying the performance bottlenecks. However, the techniques described in Section 3 (memory preallocation and automatic multi-stream execution) are well-known performance optimization tricks. Applying these tricks to squeeze the performance of DL frameworks does not have much research value, and it also makes the frameworks less general. Overall, the paper seems to be an engineering solution to a minor problem in existing DL frameworks. The idea in the paper is not a general solution that will impact the design of DL systems since it only works for static models and benefits small workload.

Correctness: Yes.

Clarity: The writing is adequate.

Relation to Prior Work: Yes.

Reproducibility: Yes

Additional Feedback: Have you compared with DL compilers such as TVM, XLA? Most of the framework overhead identified in the paper will not exist in compiled code.


Review 3

Summary and Contributions: The paper presents Nimble, a lightweight executor for static deep neural networks on GPU. Nimble achieves significantly lower overhead compared to popular neural network frameworks by 1) hoisting GPU-data-independent tasks out of execution kernels to be done once in initialization phase, and 2) automatically executing GPU operations in parallel on multiple GPU streams. Contributions 1. A lightweight static deep neural network executor for GPU. 2. Theoretical correctness proof of the algorithms. 3. Experimental results demonstrating up to 10.98x and 2.79x inference and training speedups for 13 popular networks on 3 different GPU architectures. ===Update=== I have read the author feedback and other reviews. I appreciate the additional results and the answers. (I agree the bipartite graph is needed. Thank you for the counterexample!) I've updated my score from 6 to 7.

Strengths: 1. Good speedups results against reasonable baselines, providing insights on how each optimizations (framework overhead elimination and multi-stream execution) impact the performance. 2. Demonstrates generalizability: Has results for 10+ models and 3 GPUs. 3. The algorithm is theoretically sound. 4. Innovative approach: - By limiting the scope of the framework (to static graphs and GPU-execution-only), Nimble can utilize CUDA Graphs to get seamless, low-overhead launches of the GPU kernels of the entire network. I haven't seen major frameworks really take advantage of CUDA graphs yet. (Not counting TensorRT since its internal isn't open-sourced.) - Low launch overhead allows Nimble to benefit from multi-stream execution. The algorithm to maximize the number of streams seems to be new in the deep learning context. 5. Sufficient coverage of related work. Clearly states what new contributions are. 6. Is relevance to the community. There are many static-model GPU inference/training use cases that could benefit from this work.

Weaknesses: 1. The explanation on Nimble itself is too brief. Some parts are ambiguous. Please see the Clarity section for examples. 2. Parts of the proofs are not very rigorous or organized. 3. The stream assignment algorithm could be improved / simplified. I don't think Nimble needs to construct the bipartite graph (doubling the number of nodes) and then uses max flow to solve for maximum matching. Instead, it can just augment the minimum equivalent graph G' with a source node (with edges to every root nodes in G') and a sink node (with edges from every leaf nodes in G') and then solve for max flow (all edges have weight 1). Please correct me if I'm wrong. 4. The results don't show the initialization / off-line costs. They are relatively cheap and will be amortized over iterations, but it would still be good to know. ===Update=== I take back #3. The authors gave a counterexample to show that the bipartite graph is necessary.

Correctness: Yes.

Clarity: Some parts are ambiguous or not well-defined. For example, - The stream capture part should be elaborated more. Are cudaStreamBeginCapture and cudaStreamEndCapture called only once per stream, throughout the execution of the entire graph? If so, why does figure 4 has "Intercept and Record (CUDA Stream Capture)" for each op execution? This makes it sounds like BeginCapture and EndCapture might be called at the beginning and the end of each op instead of the whole graph. - The paper should define how to count synchronizations in the graph up front, i.e., when describing the algorithm. As it is written, I could guess how it is counted, but wasn't sure until I saw the proof in the appendix. - The paragraph starting at line 239 felt rushed and could be expanded. - The minimum equivalent graph (MEG) should be defined mathematically in addition to the description "subgraph with the same set of the nodes and the smallest subset of the edges that maintains the same reachability relation as G". The paper should also mention an algorithm to derive this subgraph from G. Both of this could be achieved by citing reference [2] of the supplementary material in the paper. - Line 260 "We only report stable results in the training experiments" -- What does "stable results" mean? - Table 1: Speedup over what baseline? - Proof of Theorem A.1. is not very rigorous. - Line 53 of supplementary material: I find this paragraph a little hard to follow. After the first sentence "Here we can demonstrate that $M_f$ is a match of B." It would be helpful to add another sentence explaining what the proof is trying to do instead of just diving into the math. For example, "We will prove by contradiction. Suppose $M_f$ is not a matching and has $x_i, x_k$...".

Relation to Prior Work: Yes.

Reproducibility: Yes

Additional Feedback: Additional references that might be worth adding: - Line 89 could add [1, 2] to low-overhead languages. - Line 93 could add [3] to GPU kernel optimization works. - One related work that also focuses on lowering framework overhead is the new TensorFlow runtime [4]. (Different approach from Nimble.) [1] NVIDIA TensorRT: https://developer.nvidia.com/tensorrt [2] TensorFlow C++ API: https://www.tensorflow.org/api_docs/cc [3] Vasilache, N., Zinenko, O., Theodoridis, T., Goyal, P., DeVito, Z., Moses, W. S., ... & Cohen, A. (2018). Tensor comprehensions: Framework-agnostic high-performance machine learning abstractions. arXiv preprint arXiv:1802.04730. [4] https://blog.tensorflow.org/2020/04/tfrt-new-tensorflow-runtime.html Typos: - Line 258 "As to" -> "As for". - Line 262: "four" -> "three". - Line 123 of Supplementary Material: "becuase" -> "because".


Review 4

Summary and Contributions: The authors present Nimble, a system for fast execution of models when batch sizes are small and overhead costs (such as initialization of kernels) are the bottleneck. The authors propose to pay a one-time cost of initializing all kernels etc. (Nimble only applies to static architectures) followed by multi-stream parallel processing to speed up runtime. Experiments cover a host of networks which is nice and show that, Nimble can outperform other engines such as PyTorch, TorchScript and Caffe2. I have gone through the author response.

Strengths: Nimble can speed up (sometimes significantly) machine learning models on a wide variety of networks. The parallelization algorithm (algo 1) is a nice adaptation of previous ideas. I found the writing quality to be clear and precise.

Weaknesses: Not any obvious ones.

Correctness: As far as I can make out, the claims seem to have been clearly validated via experiments.

Clarity: Yes

Relation to Prior Work: Yes

Reproducibility: Yes

Additional Feedback:

[Author Response · NeurIPS 2020]

Thank you for the insightful comments and the opportunity to follow up.

**R1, R2, R3: Comparing Nimble with TensorRT, TVM, TensorFlow(XLA).** TensorRT and TVM employ graph
optimizations (e.g., aggressive operator fusion) and kernel selection/tuning, which are orthogonal to our idea. We
applied operator fusion (less aggressive than TensorRT's) and basic kernel selection for Conv op (use either cuDNN or
PyTorch's native implementation) to Nimble and measure its performance. Figure 1 shows that Nimble outperforms all
cases except MobileNet V2 (TVM). The reason is that TVM spends more than a day in tuning Conv kernels of a model,
and such tuning happens to be remarkably effective on MobileNet V2, finding more efficient kernels compared to those
of cuDNN and PyTorch. Note that TensorRT and TVM do not support training for now. We will add these results in our
revised paper, along with detailed discussions on related works including TorchScript and TFRT.

Figure 1: Speedup compared to TensorRT on inference work-
loads (batch size 1) using V100.

Figure 2: Speedup compared to Py-
Torch on training using V100.

Figure 3: An
example DAG.

**R2: Clarifying our contributions.** Nimble is the first work to automatically avoid framework overheads and aggres-
sively parallelize GPU kernels using multiple streams for static DL models. Nimble introduces ahead-of-time (AoT)
preparation to avoid framework overheads (details discussed below). This AoT preparation can be done quickly, and
Nimble does not experience the overheads when executing the DL model afterwards. Furthermore, this opens up an
opportunity to more efficiently utilize multiple GPU streams (as discussed in the first paragraph of §3.2). Nimble
proposes a new multi-stream algorithm that maximizes parallelism and minimizes the number of synchronizations
across streams. Nimble automates applying these techniques by capturing the GPU kernel call trace and running only
GPU kernel calls for each execution, without redesigning the framework runtime. Nimble also uses `CUDAGraph` to
reduce the number of GPU kernel launches. As a result, Nimble exhibits significant inference (and training) speedup on
various models. Moreover, Nimble is easy to use; a user just needs to wrap a PyTorch model in a Nimble object (two
additional lines), and use the Nimble object for inference or training.

Nimble's approach is different from prior approaches that identify framework overheads and remove such overheads.
Instead, Nimble captures the core DL computations (i.e., GPU kernel call trace) to run and prepares an environment for
executing the captured trace for a new input.

**R2: Sources of framework overhead.** As discussed in the paper, the framework overhead is incurred not only by
well-known sources like memory allocation but also by other sources such as inferring the output shape, dispatching
appropriate GPU kernels, and preparing GPU kernel arguments. Existing approaches (e.g., memory preallocation)
are limited to specific sources of overhead. Redesigning the framework to remove all sources is very challenging. As
described above, we present a solution to avoid all overhead sources without rewriting the framework.

**R1, R2: Large training workloads.** Figure 2 shows Nimble's performance when training larger models. We use batch
sizes of 32, 64, and 1 for BERT, ResNet50, and CycleGAN respectively. As shown in the result of BERT and ResNet50
(ImageNet), framework overhead is less pronounced when a model mostly consists of large kernels (kernels with large
amounts of computation), leading to limited performance improvement in Nimble. We will include these results to
show the limitations of Nimble in our revised paper. Nonetheless, there exist other important cases where the model
contains small kernels. For example, training classification models on the CIFAR dataset or training typical GAN
models generally involve small kernels, hence Nimble achieves training speedup on such models.

**R2: Generality of Nimble.** Nimble supports static DL models, and is not applicable to dynamic models. Yet, we
believe that Nimble covers a wide range of models and has practical, real-world impacts; for instance, TensorRT is
widely deployed in production albeit its limited applicability.

**R3: Stream capture / Initialization cost / Comments on §3.2 and Proof.** We capture all operations of the model at
once. The mean and maximum AoT preparation time for the models in Figure 1 are 0.35 s and 1.07 s (NASNet-A
(large)), respectively. We will describe the algorithm and proof more clearly in our revised paper.

**R3: Simplifying the stream assignment algorithm.** To our understanding, we cannot omit the process of constructing
a bipartite graph. We describe an example in Figure 3. Since every path from A to E includes the edge (A, B), the
maximum flow of graph is trivially 1, and does not give useful information for the stream assignment of the graph. We
greatly appreciate the suggestions.

[Meta-Review · NeurIPS 2020]

All reviewers are positive about the paper. This paper is therefore accepted. Thanks for submitting to NeurIPS and please incorporate the reviewers' comment to further improve the camera ready version.